# Parasocial Interactions in Digital Tourism: Attributes of Live Streamers and Viewer Engagement Dynamics in South Korea

**DOI:** 10.3390/bs13110953

**Published:** 2023-11-20

**Authors:** Minseong Kim

**Affiliations:** Department of Management & Marketing, College of Business, Louisiana State University Shreveport, Shreveport, LA 71115, USA; minseong.kim@lsus.edu

**Keywords:** travel live streaming, parasocial interaction, attractiveness, trustworthiness, expertise, friendship, trust, South Korean digital milieu

## Abstract

This study investigates the use of live streaming as a strategic tool in the tourism industry, with a focus on the attributes of live streamers that influence viewer engagement, particularly in the context of South Korea’s digital landscape. The purpose of this research is to understand how the attractiveness, trustworthiness, and expertise of live streamers can affect the parasocial relationships—characterized by perceived friendship and trust—between viewers and streamers and how these relationships subsequently influence viewer loyalty and cooperation intentions. A quantitative research methodology was employed, utilizing a structured online survey distributed by a leading market research agency in South Korea. The survey targeted a diverse demographic to ensure a comprehensive analysis of digital consumer behavior in the tourism sector. Data were analyzed using IBM SPSS Statistics 28.0 and IBM SPSS Amos 28.0, employing structural equation modeling to test the hypothesized relationships. The results revealed that while attractiveness and trustworthiness impact perceived friendship, only expertise significantly affects trust for a travel live streamer. Furthermore, perceived friendship plays a crucial role in fostering loyalty to and cooperation with the streamer, having practical implications for the tourism industry in terms of crafting marketing strategies and training digital ambassadors. Th study extends parasocial interaction theory to the digital travel domain, providing original insights into virtual tourist behavior and highlighting live streaming’s significant contribution to viewer engagement. This research has limitations in its geographical focus on South Korea, suggesting the need for cross-cultural studies to validate the findings. Overall, this study offers valuable contributions to the academic literature and practical guidance for the tourism industry, emphasizing the importance of digital personalities in post-pandemic tourism recovery strategies.

## 1. Introduction

In late 2019, global industries faced unparalleled disruptions due to the emergence of the COVID-19 pandemic. The tourism sector, in particular, experienced pronounced challenges [1,2]. The imposition of travel bans and lockdowns by various governments led to a precipitous decline in both international and domestic tourism, with effects lingering beyond the initial phases of the pandemic [3]. In light of these challenges, the tourism industry pivoted towards innovative solutions to sustain engagement and maintain economic viability. A salient adaptation was the use of live streaming as a medium to reach potential tourists [2,4]. The essential technology underpinning this initiative, rooted in social media platforms, is both readily available and cost-effective for rapid deployment. The travel industry, spurred by dynamic marketing strategies and virtual events, is poised for considerable economic gains. McKinsey’s projections indicate that these advances could result in revenues surpassing USD 20 billion by the year 2030. This digital modality has facilitated the virtual showcasing of tourism destinations by effectively circumventing the constraints of physical presence [2,5]. The experience provided by live streaming is both immersive and interactive, granting potential travelers the ability to virtually explore destinations from the confines of their domiciles [3,4]. Yet, the success of live streaming in the tourism sector extends beyond mere destination promotion. It is inextricably intertwined with the profile and characteristics of the individual live streamer [6]. These streamers, as the primary conduits of the experience, play an influential role in shaping viewer perceptions, desires, engagement, and consequent decision making [5,7]. Central to this study is the hypothesis that a trio of attributes—attractiveness, trustworthiness, and expertise—are pivotal in determining the impact of live-streamed tourism content [3,7,8].

Delving into the underlying psychological dynamics of viewer–streamer engagement necessitates an exploration of parasocial interaction theory [9,10,11]. Historically rooted in media studies, parasocial interaction underscores the unidirectional relationships that audiences foster with media personalities [11]. Within this framework, viewers often cultivate sentiments of intimacy, camaraderie, and trust toward such figures, despite the lack of reciprocal interaction [9,10]. Our study extrapolates this theory into the realm of travel live streaming, with a particular emphasis on the South Korean digital milieu. Our aim is to discern the nuances of perceived friendship, trust, loyalty, and cooperation, thereby revealing the magnitude of influence that travel live streamers exert over their respective cohorts.

While the extant literature abounds with research focusing on the dynamics between influencers or celebrities and their respective audiences, there remains a discernible gap in the context of travel live streamers [9]. This oversight is intriguing, especially considering the inherently experiential nature of the tourism industry, which ostensibly provides a fertile ground for rich parasocial engagements [11]. This study marks a significant theoretical advancement by applying and extending parasocial interaction theory to the emergent domain of travel live streaming [9,12]. This theory has been predominantly associated with traditional media, and its exploration within digital spaces—particularly those as rich and complex as the tourism industry—has been limited [13]. We extend this theory to travel live streaming, focusing on the psychological dynamics of viewers’ engagement with live streamers, to test and expand parasocial interaction theory. Our research challenges existing paradigms by examining the unique attributes of live streamers—specifically attractiveness, trustworthiness, and expertise—and their influence on the intensity and nature of parasocial relationships. This study deepens our theoretical understanding of the intersection between digital and physical interactions, offering insights into how these dimensions coalesce to shape viewer perceptions in digital environments [14,15,16].

In addition, given South Korea’s leading role in global digital trends and the explosive international interest in Korean culture—propelled by the Hallyu wave—the insights gleaned from this study are likely to have far-reaching implications [17]. They have the potential to redefine theoretical assumptions about digital engagement, extending well beyond the Korean peninsula [17,18]. This research capitalizes on the opportunity to scrutinize how cultural specificity and technological advancement intersect to shape new paradigms of viewer–streamer interactions, thereby enhancing the depth and breadth of parasocial interaction theory. The overarching objective is thus not merely to bridge an existing academic void but to proffer insights germane to a pivotal player in Asia’s digital arena.

For practitioners within the tourism industry, the implications of this research are both strategic and operational. By identifying key streamer attributes, our study offers a blueprint for the selection and training of digital ambassadors capable of maximizing viewer engagement and, by extension, potential tourism revenue. Furthermore, it guides industry stakeholders in crafting targeted marketing campaigns that leverage these insights to attain a competitive advantage. In the context of South Korea’s robust digital infrastructure and innovative marketing landscape, these insights are particularly pertinent, offering a model for best practices that can be adapted across various digital platforms and cultural milieus [17,18]. As the digital transformation of the tourism industry accelerates, our research serves as a critical touchstone for understanding the evolving nature of consumer engagement in the digital age. This is not merely an academic exercise; it is an essential exploration of the future of tourism and digital interaction. The urgency and relevance of this study are underscored by the ongoing impacts of the pandemic, and the imperative that tourism industries must adapt swiftly and effectively to the new normal that will define the post-pandemic world.

In conclusion, as the digital realm continues to reshape the contours of the tourism industry, this research stands at the nexus between theoretical innovation and pragmatic application. By dissecting the intricate dynamics of travel live streaming in the South Korean landscape, we intend for our research to be a beacon for both scholars and practitioners, guiding them through the multifaceted challenges and opportunities of the digital age. 

The structure of this article encompasses the following components: An introduction section (Section 1), which establishes the research context, the foundational premise, and the scholarly significance and contributions of this research. Next is the Literature Review and Hypothesis Development section (Section 2), wherein existing academic discourse is systemically examined, and specific research hypotheses are theoretically articulated. The Methods section (Section 3) is meticulously divided into three sub-components—data collection, measures, and data analysis—each delineating the intricate methodological strategies employed in this study. After the methodology is described, the results are articulated through measurement model and structural model assessments, offering a rigorous evaluation of this study’s data. The article culminates with the Conclusions and Implications section (Section 5), which is subdivided into Theoretical Implications, providing an academic contribution to the field; Managerial Implications, delivering actionable insights for industry practitioners; and finally Limitations and Directions for Future Research, which presents a critical reflection on the study’s constraints and suggests prospective research trajectories.

## 2. Literature Review and Hypotheses Development

This research model presents a novel contribution to the extant literature on tourism and the parasocial interaction framework by meticulously dissecting a triad of live streamer attributes—attractiveness, trustworthiness, and expertise—and their mediated influence, through perceived friendship and trust, on viewer loyalty and cooperation. This nuanced approach not only enriches the traditional parasocial interaction framework with a contemporary digital twist but also foregrounds the nuanced mechanics of engagement in the virtual tourism domain [12,14,19]. By integrating mediators that encapsulate the essence of parasocial relationships within the live streaming context, this model deepens our understanding of how viewers form bonds with digital personalities, thereby providing a more granular view of consumer behavior [12,20,21]. This granular view is particularly salient in the tourism literature, where such dynamics have been underexplored, especially in relation to the burgeoning practice of virtual tourism via live streaming [13,15]. The inclusion of loyalty and cooperation as outcomes offers a pragmatic angle of digital engagement, aligning theoretical insights with actionable metrics pertinent to scholars and practitioners in the digital consumer behavior field [22]. Consequently, our research model stands out by bridging a gap in digital tourism studies, positioning the parasocial interaction framework as a pivotal lens through which the impact of travel live streamer attributes on consumer engagement can be comprehensively understood and leveraged in the wake of the digital transformation of tourism experiences (see Figure 1).

Parasocial interaction refers to the one-sided relationships that audience members form with media figures [14,23,24]. Viewers—in this case, those watching a live stream—develop feelings and attachments similar to those in real-life social interactions, even though the media figures (e.g., live streamers) do not reciprocate these feelings individually [25,26,27]. From a psychological standpoint, *attractiveness* is not just about physical appearance [12,15,23]. It encompasses a range of qualities that make an individual appealing to others. In the context of parasocial interaction, when viewers find a live streamer attractive, they are more likely to invest emotionally in the content, leading to a deeper perceived connection or friendship [26]. The “halo effect”—a cognitive bias wherein our impression of someone in one area (e.g., physical attractiveness) influences our overall impression of that person—can make viewers ascribe other positive traits to the streamer, strengthening the sense of friendship [13,19,25,28].

Trust is foundational in the realm of live streaming. *Trustworthiness* in this setting relates to a viewer’s belief that a streamer is genuine, consistent, and reliable [27]. Psychologically, when individuals deem someone trustworthy, they are more likely to share personal experiences and emotions and invest time in the relationship [29]. In a parasocial context, this means that viewers are more likely to return to the streamer’s broadcasts, engage more deeply in their content, and perceive a stronger friendship with the streamer [25,28]. *Expertise* is a crucial factor in determining how viewers evaluate the credibility and value of a live streamer’s content [24,29]. From a psychological viewpoint, expertise creates a sense of respect and admiration. When viewers believe that a streamer possesses knowledge or skills that they value, they are more likely to form a deeper parasocial bond [29]. This bond can be perceived as a form of friendship, where the viewer tunes in not just for entertainment but also for learning, guidance, or advice [24,27]. Therefore, the following hypotheses were developed:

**H1-1.** *Attractiveness is positively associated with perceived friendship*.

**H1-2.** *Trustworthiness is positively associated with perceived friendship*.

**H1-3.** *Expertise is positively associated with perceived friendship*.

The impact of a live streamer’s attractiveness, trustworthiness, and expertise on viewer trust is multifaceted, with each factor playing a significant role in shaping perceptions. Specifically, the halo effect suggests that people instinctively place more trust in individuals they find attractive, even if that trust is not necessarily rooted in reason [30,31]. Therefore, live streamers who are deemed visually appealing not only captivate their audiences more effectively but are also often ascribed attributes like sociability, humility, and kindness [31,32]. This positive perception extends to viewers associating themselves with traits like knowledgeability and determination. The underlying mechanism, as supported by Filieri et al. [30], emphasizes that a streamer’s physical allure plays a vital role in fostering viewer trust, highlighting the profound influence of attractiveness on viewer engagement.

Likewise, a sense of trustworthiness emerges as a foundational pillar in establishing trust. Under ideal conditions, trust is a byproduct of perceived trustworthiness [31]. Empirical evidence from the professional sphere, as presented by Gabrieli, Ng, and Esposito [33], underscores this dynamic relationship: individuals exhibit trust when they perceive trustworthiness. Extending this to the realm of live streaming, viewers who recognize a streamer’s credibility are more likely to engage deeply with the content, solidifying their relationship through continuous interaction [18,33]. Trustworthiness therefore serves as a reliable predictor of the depth of trust that an individual can inspire [30].

Lastly, expertise has a unique influence on building trust. Drawing from social exchange theory, Kim and Kim [18] found that an influencer’s expertise acts as a catalyst for followers’ emotional trust. Delving deeper into the live streaming milieu, Zhang et al. [34] confirmed that viewers’ trust is significantly influenced by their perception of a streamer’s knowledge and proficiency. This expertise not only enhances content receptivity but also steers viewers towards positive interactions, leading them to regard these streamers as genuinely trustworthy entities [35]. In specialized streams like travel, expertise is doubly crucial. Here, the streamer often functions as both a tour guide and a seller. Historical data emphasize that expertise, whether in guiding tourists or in sales, is a cornerstone of building trust [18,30,34]. Collectively, this suggests the following: 

**H2-1.** *Attractiveness is positively associated with trust in a travel live streamer*.

**H2-2.** *Trustworthiness is positively associated with trust in a travel live streamer*.

**H2-3.** *Expertise is positively associated with trust in a travel live streamer*.

The landscape of social media has provided fertile ground for nurturing perceived friendships, which, in turn, play a profound role in shaping psychological outcomes related to trust, loyalty, and cooperation among users [36]. First, at a cognitive level, perceived friendships can create a psychological safety net, fostering a belief in the authenticity and reliability of online personalities [37]. When users feel they share a “friendship” with online entities, they are more inclined to trust them, as these relationships often mimic the trust dynamics present in offline friendships [36,37]. On an emotional level, perceived friendships can elicit a sense of belonging and attachment, reinforcing loyalty [18,38]. Users who believe they share a bond or friendship with online personalities or brands may develop an emotional investment in them, which transcends typical user-entity relationships [35]. This emotional connection acts as a catalyst for sustained loyalty, as individuals tend to remain loyal to those to whom they feel emotionally connected [37,38]. Behaviorally, perceived friendships act as motivators, nudging users towards collaborative actions [22,35]. In essence, when users perceive a sense of mutual friendship, they are more likely to engage in cooperative behaviors, stemming from a psychological desire to contribute positively to the friendship [36]. Consequently, this gives rise to the following hypotheses:

**H3-1.** *Perceived friendship is positively associated with trust in a travel live streamer*.

**H3-2.** *Perceived friendship is positively associated with loyalty*.

**H3-3.** *Perceived friendship is positively associated with cooperation*.

Previous research has emphasized the compelling connection between trust and both loyalty and cooperation in the digital realm [39,40,41]. When consumers place trust in a service provider, this establishes the foundation for enhanced loyalty and increased cooperation. This trust is instrumental in fostering long-standing associations and encouraging consumers to navigate the uncertainties that come with such affiliations—thus playing an essential role in cultivating mutual value [39,40,41,42]. Accordingly, heightened online trust translates into varied cooperative actions of consumers [39,41], encompassing both active involvement and broader community contributions. When there is a prevailing trust sentiment, consumers exhibit a stronger inclination toward engagement in online platforms [40,42]. This means that members of digital communities are more likely to exchange information, build meaningful connections, and actively participate in discussions, given their robust trust in the platform. Thus, the following hypotheses were formed:

**H4-1.** *Trust in a travel live streamer is positively associated with loyalty*.

**H4-2.** *Trust in a travel live streamer is positively associated with cooperation*.

## 3. Method

### 3.1. Data Collection

Data collection was orchestrated in collaboration with a prominent online market research agency in South Korea during the month of April 2023. Esteemed for its expansive reach within the national digital sphere, this agency boasts a sophisticated online survey framework, granting access to a comprehensive and diverse respondent pool. This extensive database is inclusive of myriad enterprises, spanning a range of sectors, and it notably incorporates associations with the vast majority of South Korea’s higher-education institutions. To ensure the integrity and credibility of the survey responses, the platform integrates meticulous verification mechanisms. The tracking of IP addresses is employed to affirm the uniqueness of each respondent’s submission. Additionally, the platform is equipped to detect and scrutinize outlier responses, typically characterized by notably brief or protracted completion intervals. Prior to the disbursement of any incentives associated with survey participation, respondents are subjected to a stringent identity authentication process.

For the survey rollout, a preliminary pool of 2000 participants was indiscriminately selected, via a random sampling approach, to address an initial inquiry: “Over the past 3 months, have you engaged with travel-centric live streaming content?” From this cohort, 610 respondents satisfied this criterion and proceeded to complete the entirety of the survey. After rigorous data scrutiny, in which responses with inconsistencies such as random-pattern answers and atypical completion durations were eliminated, a consolidated total of 575 responses were earmarked for advanced multivariate analysis. The demographic characteristics of the participants in this study are shown in Table 1.

### 3.2. Measures

For this study, multi-item scales were meticulously sourced from established academic literature, selected primarily for their empirical rigor as well as demonstrable reliability and validity in prior applications. Tailoring them to the unique parameters and thematic underpinnings of our research, certain items were subjected to subtle refinements to ensure contextual congruence. Specifically, constructs like expertise, trustworthiness, and attractiveness were articulated through four items, drawing inspiration from the academic work of Li and Peng [43]. Our examination of the perceived friendship construct drew from seven items adapted from Kim and Kim [35], while the trust construct integrated three items anchored in Wongkitrungrueng and Assarut [44]’s research. Constructs related to loyalty and cooperation found empirical grounding in the comprehensive work of Kim and Kim [22].

Respondents assessed each item using a 7-point Likert scale, spanning from 1 (strong disagreement) to 7 (strong agreement). Given the crucial nature of clarity and user comprehensibility in survey design, we piloted this instrument using a sample of 77 individuals who had previously engaged in travel live streaming within the past month. The invaluable feedback received during this pilot phase allowed us to fine-tune any ambiguously framed items, thereby optimizing the survey’s coherence. In a bid to mitigate potential pitfalls associated with common method bias—a concern inherent in survey-based research—our study instituted a procedural countermeasure [45,46]. Specifically, survey items were presented in a randomized sequence for each respondent. This methodological choice was anchored in the rationale that randomized item presentation disrupts response patterns linked to methodological artifacts, thus serving as a buffer against the possible distortions of common method bias [45,46].

### 3.3. Data Analysis

In this study, quantitative analyses were conducted using IBM SPSS Statistics 28.0 and IBM SPSS Amos 28.0 to ensure methodological precision. Frequency analysis via SPSS was employed to delineate the distribution of demographic variables, providing foundational insights into the sample characteristics. Correlation analysis was also undertaken to ascertain discriminant validity, thereby confirming that the study constructs were measured with sufficient distinction and minimal collinearity. Further methodological robustness was achieved through the deployment of IBM SPSS Amos 28.0, which was instrumental in performing confirmatory factor analysis and structural equation modeling. The confirmatory factor analysis was meticulously conducted to substantiate convergent validity, with each construct demonstrating satisfactory standardized estimates, critical ratios, composite reliability scores, and average variance extracted values. Notably, discriminant validity was rigorously verified; the average variance extracted values for each construct were greater than the corresponding squared inter-construct correlation coefficients, thus reinforcing the constructs’ uniqueness. Subsequently, structural equation modeling was executed to elucidate the interrelationships among the constructs and rigorously test the proposed theoretical framework underpinning the hypotheses.

## 4. Results

### 4.1. Measurement Model Assessment

The preliminary step involved assessing the measurement model using confirmatory factor analysis (CFA) to substantiate the relationships between the observed variables (or indicators) and their corresponding latent constructs. The fit indices from the CFA indicated a satisfactory alignment with the data. Specifically, the model returned the following metrics: chi-square = 978.274, degrees of freedom = 382, *p* < 0.001, chi-square/degrees of freedom = 2.561 (i.e., indicating a good fit, as it is below the 3.0 threshold), Root Mean Square Error Approximation (RMSEA) = 0.052 (i.e., signifying a close fit, since it is under 0.80), Comparative Fit Index (CFI) = 0.940, Tucker-Lewis Index (TLI) = 0.932, and Normed Fit Index (NFI) = 0.906—all of which surpass the benchmark value of 0.90 [47]. Additionally, all indicators exhibited standardized estimates and critical ratios that surpassed the widely accepted thresholds in social sciences, with values greater than 0.40 and 1.96, respectively (*p* < 0.05) [47] (see Table 2).

Reliability analyses revealed that there was strong internal consistency for all the constructs. The composite reliability (CR) values ranged from 0.746 (trust in a travel live streamer) to 0.940 (perceived friendship), well above the recommended threshold of 0.70 [47] (see Table 3). Convergent validity was confirmed since the average variance extracted (AVE) values for all the constructs exceeded the 0.50 benchmark, with values spanning from 0.502 (trust in a travel live streamer) to 0.758 (perceived friendship). Furthermore, to assess discriminant validity, the square root of each construct’s AVE was compared with its correlations with other constructs. As anticipated, the square root of the AVE for each construct was greater than its highest correlation with any other construct, thus confirming discriminant validity (see Table 3).

### 4.2. Structural Model Assessment

The structural model exhibited a satisfactory fit with the data: chi-square = 1059.165, degrees of freedom = 388, *p* < 0.001, chi-square/degrees of freedom = 2.730, RMSEA = 0.055, CFI = 0.933, TLI = 0.925, and NFI = 0.900. The squared multiple correlations revealed that the hypothesized model accounted for 30.1% of the variance in perceived friendship, 19.5% of the variance in trust in a travel live streamer, 35.7% of the variance in loyalty, and 36.8% of the variance in cooperation. 

The empirical results of structural equation modeling revealed that perceived friendship was significantly influenced by attractiveness (standardized estimate = 0.188, critical ratio = 3.568, standardized error = 0.067, and *p* < 0.01), trustworthiness (standardized estimate = 0.197, critical ratio = 3.169, standardized error = 0.068, and *p* < 0.01), and expertise (standardized estimate = 0.277, critical ratio, 5.137, standardized error = 0.057, and *p* < 0.01), thus supporting H1-1, H1-2, and H1-3. However, the empirical findings revealed that trust in a travel live streamer was significantly affected by expertise (standardized estimate = 0.307, critical ratio = 4.576, standardized error = 0.067, and *p* < 0.01), while attractiveness (standardized estimate = 0.064, critical ratio = 1.025, standardized error = 0.076, and *p* > 0.05) and trustworthiness (standardized estimate = 0.048, critical ratio = 0.652, standardized error = 0.077, and *p* > 0.05) did not have a significant effect on trust in a travel live streamer, thereby supporting H2-3 only.

The empirical outcomes demonstrated that perceived friendship had statistically significant influences on trust in a travel live streamer (standardized estimate = 0.115, critical ratio = 2.065, standardized error = 0.053, and *p* < 0.05), loyalty (standardized estimate = 0.424, critical ratio = 8.245, standardized error = 0.035, and *p* < 0.01), and cooperation (standardized estimate = 0.506, critical ratio = 10.893, standardized error = 0.043, and *p* < 0.01), hence supporting H3-1, H3-2, and H3-3. Lastly, the empirical findings indicated that trust in a travel live streamer significantly influenced loyalty (standardized estimate = 0.310, critical ratio = 5.766, standardized error = 0.038, and *p* < 0.01) and cooperation (standardized estimate = 0.213, critical ratio = 4.549, standardized error = 0.045, and *p* < 0.01), thus supporting H4-1 and H4-2 (see Table 4).

## 5. Conclusions and Implications

### 5.1. Theoretical Implications

Building upon Horton and Wohl’s [48] foundational work on parasocial interaction, this study introduces a pioneering exploration of the theory within the context of digital tourism, particularly through the lens of live-streaming services in South Korea. Unlike prior studies that have primarily focused on traditional broadcast media, this research delves into the nuances of live streaming as a novel medium for tourism promotion [9,10,11]. By doing so, it transcends traditional media boundaries within the tourism field and examines how real-time interaction and viewer engagement with travel live streamers can mimic and potentially exceed the depth of parasocial relationships previously observed in the context of television and radio.

This study distinguishes itself by highlighting expertise as the central facilitator of parasocial interaction, unlike earlier research that emphasized attractiveness and trustworthiness [49,50]. Through this focus, this research reveals that, in the context of live streaming for tourism, the informative value and knowledge provided by a streamer are paramount. This is a significant shift from the earlier literature, where the appeal of a personality often took precedence over content quality [51,52]. The South Korean context, known for its advanced digital culture, underscores this finding, which suggests that digitally savvy audiences are more discerning, seeking substantial and expertly presented tourism content [18,53].

Further advancing the parasocial interaction framework, this study identifies the roles that perceived friendship and trust play in mediating the relationship between live streamer attributes and viewer behavioral intentions (such as loyalty and cooperation). This intricate relationship is a novel contribution to the extant literature, showcasing a complex psychological interplay that has not been extensively examined in the context of tourism live streaming [21,54]. The empirical evidence provided by this study indicates that these mediating factors are not peripheral but central to the effectiveness of live-streaming services in fostering viewer engagement and conversion [37,54].

A unique angle of this research is the prioritization of credibility over personality within the realm of parasocial relationships. The findings indicate that in the livestreaming tourism domain, viewers prioritize the streamer’s expertise over other attributes, suggesting a more rational and discerning approach to forming parasocial relationships [50,51,52]. This study challenges the extant paradigms that have often represented the viewer–streamer bond as being primarily influenced by charisma or attractiveness, thus advancing the conversation toward intellectual engagement between the streamer and the viewer.

In conclusion, this research contributes to a deeper theoretical understanding of digital consumer behavior within the tourism industry. It particularly emphasizes how digital platforms, such as live streaming, can engender relationships that are rich, multi-dimensional, and reflective of genuine social interactions [21,55]. By doing so, it extends the applicability of parasocial interaction theory to digital consumer behaviors, especially within the unique cultural and technological landscape of South Korea—a region at the forefront of the Hallyu wave and digital innovation [17,26]. This study provides a model for understanding the impact of live streamer attributes on engagement and behavior in digital tourism contexts across Asia. These insights necessitate a reevaluation of parasocial interaction theory in the context of digital transformation, suggesting new directions for future research in the rapidly evolving digital tourism landscape, with a particular focus on the Asian market.

### 5.2. Managerial Implications

When selecting live streamers, tourism organizations should prioritize a mix of charismatic presentation and authoritative knowledge. For instance, a regional tourism board could collaborate with well-known local chefs who can offer culinary tours through live streams, combining their attractiveness and culinary expertise to engage viewers. Training programs could be modeled after professional broadcasting workshops, where streamers practice various scenarios, from dealing with technical difficulties to handling live audience questions with poise and depth. These sessions could be recorded and reviewed, akin to sports teams analyzing game tapes, allowing streamers to refine their approaches. Streamers could also be trained in leveraging the storytelling techniques used in documentaries in order to make historical sites come alive for their viewers.

Both destination marketing managers and live streamers should consider employing real-time analytics tools that can measure viewer engagement, similar to television ratings systems, during live streams. These data can help in making immediate content adjustments, much like how live broadcasters gauge audience reactions. For a longer-term strategy, destination marketing managers and streamers can examine viewer retention data to determine which parts of a stream are most engaging. For example, if viewers drop off when a streamer moves away from interactive content, future live-streaming services and content could be adjusted to include more question-and-answer (Q&A) segments, live demonstrations, or virtual “hands-on” experiences. Interactive experiences could be taken a step further by integrating them into a broader loyalty program. For example, viewers could earn points not only for watching and interacting with live streams but also for participating in post-stream surveys, one-time events/deals, or contributing to community-driven content (such as suggesting future stream topics or destinations). These points or benefits could then be used in a tourism organization’s wider ecosystem, perhaps providing discounts on partnered hotels or exclusive access to behind-the-scenes content from popular streamers, similar to the reward systems prevalent in gaming communities.

Embracing cutting-edge technology, tourism organization managers could launch a digital experience-centered technology that is paired with live streams, offering viewers the ability to place themselves within the streamed environment via a technological interface, thereby enhancing the sense of presence. Imagine a live stream from a bustling street market, where viewers can use an app to virtually “try on” traditional clothing or “sample” street food. For this technology, tourism boards could create a series of immersive experiences tied to the live stream, allowing viewers to explore a destination before the live event—like a “pre-tour”—and maintain engagement with post-stream digital experiences that delve deeper into the highlighted locations. Trust can also be fostered through consistent and transparent communication strategies. Live streamers could have regular “office hours” where viewers can join live video chats to discuss travel plans and concerns, replicating the accessibility of a trusted travel agent. Travel live streamers could also hold live “reaction” sessions where they watch and discuss recent travel news or trends, providing a platform where viewers can feel that they are part of an informed community. This approach would mirror successful strategies used by influencers and media personalities who maintain transparency and open dialogue with their followers.

The comprehensive approach for a tourism campaign should be holistic, integrating each of these detailed strategies into a seamless operation. The campaign could be unveiled with a grand kickoff event that showcases the selected streamers, introduces the interactive app, and schedules upcoming streams. This could be followed by a season of streaming content, where each episode builds on the previous one, encouraging viewers to follow along and participate in an evolving narrative. This serial approach, much like a television series, would keep viewers engaged over a longer period, with each stream offering new information, interaction, and technology integration.

### 5.3. Limitations and Directions for Future Research

Our research undeniably makes a significant contribution to the existing literature, furnishing both academics and industry experts with invaluable insights into the viewer–streamer dynamic. However, it is imperative to acknowledge some inherent limitations. The primary concentration on South Korea’s digital landscape, sophisticated though it may be, may not wholly encapsulate global patterns in viewer–streamer relationships. The cultural idiosyncrasies and technological peculiarities inherent to the Korean context could color the discerned relationships, thus tempering the direct applicability of our results in disparate sociocultural settings.

Furthermore, while the attributes scrutinized in this study have proven to be impactful, they represent a subset of potential determinants, shaped by both conceptual deliberations and empirical constraints specific to this research. Other salient factors could also be instrumental in modulating viewer–streamer interactions. As such, subsequent research endeavors would benefit from adopting a more diversified methodological stance, perhaps integrating qualitative techniques (like comprehensive interviews) or employing advanced data-gathering methods (such as online review mining).

Looking forward, it would be judicious for future inquiries to validate and, if necessary, refine our findings across varied geographical and cultural landscapes. This would facilitate a more nuanced and holistic view of the phenomenon. Delving into other potential attributes and gauging their respective influences on constructs like perceived friendship and trust could offer more comprehensive insights. Moreover, in an age characterized by rapid technological innovations, it is essential for scholarship to evolve in tandem, consistently evaluating the ramifications of these advancements for viewer–streamer engagements and, by extension, the tourism sector at large.

## Figures and Tables

**Figure 1 behavsci-13-00953-f001:**
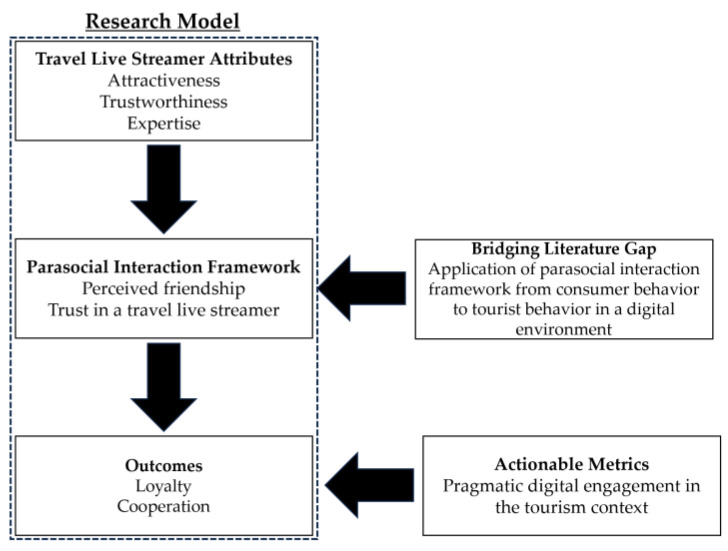
The research model, with theoretical frameworks and contributions.

**Table 1 behavsci-13-00953-t001:** Demographic analysis of respondents.

Demographic Variables		*N* = 575	Percent (%)
Gender	Female	288	50.1%
	Male	287	49.9%
Age	20s	327	56.8%
	30s	207	36.0%
	40s	35	6.1%
	Over 50s	6	1.1%
Education	High school graduate	103	17.9%
	Working on or completed associate degree	173	30.1%
	Working on or completed bachelor’s degree	274	47.6%
	Working on or completed graduate degree	25	4.4%
Education	High school	78	20.4%
	College or university degree	270	70.9%
	Graduate degree	33	8.7%
Occupation	Self-employed	110	19.1%
	Employed	258	44.7%
	Out of work and looking for work	29	5.0%
	Homemaker	65	11.3%
	Student	106	18.4%
	Military	7	1.5%

**Table 2 behavsci-13-00953-t002:** Measurement model from CFA.

Constructs and Items	Standardized Estimates	Critical Ratios
*Attractiveness*		
I feel that the travel live streamer is classy.	0.680	Fixed
I feel that the travel live streamer is handsome/beautiful.	0.830	16.856
I feel that the travel live streamer is elegant.	0.790	16.273
I feel that the travel live streamer is sexy.	0.778	16.076
*Trustworthiness*		
I feel that the travel live streamer is dependable.	0.755	Fixed
I feel that the travel live streamer is honest.	0.788	18.341
I feel that the travel live streamer is reliable.	0.820	19.030
I feel that the travel live streamer is sincere.	0.693	16.068
*Expertise*		
I feel that the travel live streamer is an expert.	0.809	Fixed
I feel that the travel live streamer has experience in live streaming.	0.815	21.211
I feel that the travel live streamer is knowledgeable in the field of live streaming.	0.835	21.800
I feel that the travel live streamer is qualified to broadcast live-streams.	0.735	18.658
*Perceived friendship*		
The interactions I have with the travel live streamer feel personally connected.	0.787	Fixed
I experience a sense of closeness and intimacy when I engage with the travel live streamer on his/her social media profile.	0.912	25.514
Interacting with the travel live streamer is akin to conversing with one of my close friends.	0.908	25.359
I perceive the travel live streamer as someone who could be a friend in real life.	0.874	24.065
I feel emotionally connected to the travel live streamer in real life.	0.867	23.817
I have a desire to reciprocate and support the travel live streamer in real life.	-	-
I am inclined to share my genuine thoughts and feelings with the travel live streamer.	-	-
*Trust in a travel live streamer*		
I believe in the information that the travel live streamer provides through live streaming.	0.661	Fixed
I can trust the travel live streamer when live streaming.	0.862	12.368
I do not think that the travel live streamer would take advantage of me while live streaming.	0.571	11.401
*Loyalty*		
I am inclined to favor this travel live streamer over others in the future.	0.857	Fixed
I actively seek out this streaming community for information and recommendations related to this travel live streamer.	0.869	20.541
I encourage friends and relatives to consider travel recommendations related to this travel live streamer.	0.643	12.623
I speak highly of this travel live streamer to others.	0.446	9.153
I intend to purchase products associated with this travel live streamer.	0.718	13.723
I prefer this travel live streamer over other streamers.	0.649	12.737
*Cooperation*		
I make efforts to help maintain the integrity of this streamer’s community (e.g., I report inappropriate advertisements or spam).	0.832	Fixed
I offer my full cooperation with other members of this streamer’s community.	0.795	21.036
I diligently adhere to the guidelines and policies of this streamer’s community.	0.760	19.865
I make a conscious effort to treat the moderators of and contributors to this streamer’s community with kindness and respect.	0.728	18.794

**Table 3 behavsci-13-00953-t003:** Construct intercorrelations (*Φ*), mean, SD (standard deviation), AVE (average variance extracted), and CR (construct reliability).

	AT	TW	EP	PF	TT	LT	CP
AT	1						
TW	0.496 **	1					
EP	0.358 **	0.528 **	1				
PF	0.377 **	0.422 **	0.433 **	1			
TT	0.201 **	0.256 **	0.336 **	0.286 **	1		
LT	0.296 **	0.327 **	0.459 **	0.425 **	0.329 **	1	
CP	0.279 **	0.450 **	0.459 **	0.514 **	0.303 **	0.577 **	1
Mean	4.766	5.188	5.030	4.867	4.714	5.009	5.208
SD	1.027	0.979	1.063	1.209	1.093	0.829	0.934
AVE	0.595	0.586	0.693	0.758	0.502	0.506	0.608
CR	0.854	0.849	0.876	0.940	0.746	0.855	0.861

Note. AT: Attractiveness, TW: Trustworthiness, EP: Expertise, PF: Perceived friendship, TT: Trust in a travel live streamer, LT: Loyalty, CP: Cooperation. ** *p* < 0.01.

**Table 4 behavsci-13-00953-t004:** Standardized structural estimates.

	Path	Standardized Estimate	Standardized Error	Critical Ratio	*p*-Value
H1-1	Attractiveness → Perceived friendship	0.188	0.067	3.568 **	0.001
H1-2	Trustworthiness → Perceived friendship	0.197	0.068	3.169 **	0.002
H1-3	Expertise → Perceived friendship	0.277	0.057	5.137 **	0.001
H2-1	Attractiveness → Trust in a travel live streamer	0.064	0.076	1.025	0.305
H2-2	Trustworthiness → Trust in a travel live streamer	0.048	0.077	0.652	0.514
H2-3	Expertise → Trust in a travel live streamer	0.307	0.067	4.576 **	0.001
H3-1	Perceived friendship → Trust in a travel live streamer	0.115	0.053	2.065 *	0.039
H3-2	Perceived friendship → Loyalty	0.424	0.035	8.245 **	0.001
H3-3	Perceived friendship → Cooperation	0.506	0.043	10.893 **	0.001
H4-1	Trust in a travel live streamer → Loyalty	0.310	0.038	5.766 **	0.001
H4-2	Trust in a travel live streamer → Cooperation	0.213	0.045	4.549 **	0.001

** *p* < 0.01; * *p* < 0.05.

## Data Availability

The data presented in this study are available on request from the corresponding author. The data are not publicly available for privacy reasons.

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
