# Peer review of "Parasocial Interactions in Digital Tourism: Attributes of Live Streamers and Viewer Engagement Dynamics in South Korea"

_behavsci, 2023, doi:10.3390/bs13110953_

Round 1

Reviewer 1 Report

Comments and Suggestions for Authors

I have outlined some key issues that need to be addressed for the paper to meet the standards of the journal and provide a clear contribution to the field. Please find my feedback and recommendations below:

 The paper lacks a clear theoretical or conceptual framework, which is essential for guiding the research and helping readers understand the context and motivation behind the study. I recommend that you provide a well-defined theoretical framework (with appropriate diagram).

 The paper does not sufficiently describe the methodological choices made, including the software or method used for data analysis(SPSS?/ AMOS?/ PLS etc.).

Table 4 should include p-values, as they are a fundamental component in statistical analysis.

 I kindly request that you provide a Turnitin report and an AI similarity report with your submission.

Comments on the Quality of English Language

The use of English in the paper may pose a barrier for readers. To improve accessibility and understanding, it is advisable to use simpler language throughout the paper to make the research more comprehensible and engaging for a broader audience.

Author Response

Please find the attached response letter.

Reviewer 2 Report

Comments and Suggestions for Authors

General comments:

This article focuses on the streamer's viewer engagement in the sphere of travel broadcasts in the period after the Covid-19 pandemic. Its biggest problem is the lack of comparison (research) with the time before and during the pandemic, so the study only from 2023 is difficult to analyze despite the rich mathematical and statistical apparatus. The post-pandemic period around the world was generally characterized by a sudden return to traditional tourism, which was most felt by airline companies that were unable to keep up with the transportation of willing passengers. Writing about the success of live streaming in this context is incomplete because it does not provide any effects, e.g. in the form of a positive correlation with an increased number of tourist trips. As a new marketing method, is it really universal in every culture and can the above-mentioned features actually influence the increase in sales of tourist services? The presented study only contains a model for identifying potential factors that may influence the relationship between the viewer and the moderator and performer of live streaming marketing in any field, not necessarily in the tourism industry. This dissonance should be justified.

Detailed notes:

- the Abstract should include a definition of the subject, purpose of the study, description of the method, summary of results and conclusions, research implications and limitations, originality, value and contribution of the article. Therefore, there is no clearly formulated goal of the work, the description of the method, its true value and contribution to science should be formulated more clearly.

- the Introduction should include a presentation of the research problem, the state of current research on the research problem, identification of a research gap, presentation of the objectives of the article/research questions/research hypotheses. Please complete and omit irrelevant content. The Introduction does not describe the structure of the article,

- in general, it also seems that assigning such emotionally significant and highly private characteristics as friendship, trust, loyalty and cooperation to live streaming "production" even during the Covid-19 pandemic is not justified - the experience of tourism companies in Europe has shown that nothing will replace the real, traditional form of travel. Maybe this approach is appropriate for Korea (or the PRC or the USA - where live streaming is gaining significant popularity), but is it appropriate for other cultures? Therefore, I do not think that ...this study could serve as a template for similar studies in different geographical or cultural contexts...,

- this is still a description of para-social relationships, not social ones. They are a substitute for real social relationships and may work in a situation of threat (illness, lack of finances, lack of sufficient knowledge), but not always in situations such as before and after the pandemic.

- of course, it is also necessary for people whose knowledge about the world is minimal, in this situation it can work as a good marketing method. However, the article does not give us the answer to this question.

- it focuses strictly on the analyzed phenomenon and this is its undoubted advantage,

- all statistical analyzes are also correct, whatever they prove in the above context.

- he article is compact and well placed in the literature, however, what is missing in the references is a broader perspective on this problem

Author Response

(The authors gave the same response as above.)

Reviewer 3 Report

Comments and Suggestions for Authors

Thanks for the well-written paper.  The paper selected an interesting topic.  This paper has potential if the revise can address major/minor suggestions.

This research should find a stronger argument regarding (1) why the study is important and (2) why the findings are meaningful for scholars/practitioners.  The introduction section should be strengthened.

In the literature review section, when the author(s) explain parasocial interaction, it is necessary to provide more recent references.  For instance, the below references can be heplful.

Chen, X., Hyun, S. S., & Lee, T. J. (2022).  The effects of parasocial interaction, authenticity, and selfcongruity on the formation of consumer trust in online travel agencies.  International Journal of Tourism Research, 24(4), 563-576.

The discussion is a little disappointing and could be written in a much stronger way.  Also, there are very little managerial implications, and more specific, poignant recommendations should be provided to the practitioners/managers.  This information would give the paper a much better finish.

Author Response

(The authors gave the same response as above.)

Round 2

Reviewer 1 Report

Comments and Suggestions for Authors

I express my gratitude to the author(s) for their meticulous review of my feedback and making necessary changes.

One enhancement I'd appreciate is the inclusion of a path diagram illustrating the hypothesised causal relationship.

Author Response

Thank you for your kind words and positive evaluation of my revised manuscript. I am pleased to hear that my revisions have met your expectations, and I appreciate your thorough review of our work.

Regarding your suggestion to include a path diagram to illustrate the hypothesized causal relationships, I find this to be a valuable addition. A path diagram can indeed enhance the clarity of our proposed model by visually representing the relationships among variables.

Once again, I am grateful for your insightful feedback and support in enhancing the quality of our work.

Best regards,

Minseong Kim

Reviewer 3 Report

Comments and Suggestions for Authors

The author well-revised the paper based on the referee's suggestions.

Thus, I accept this paper. 

Author Response

Dear Reviewer 3,

I am sincerely grateful for your positive evaluation and the acceptance of my revised manuscript. It is heartening to know that the revisions made in response to your suggestions have been well-received.

Your constructive guidance throughout the review process has been invaluable in enhancing the quality and clarity of my work. I am delighted to see that my efforts have aligned well with your expectations and the standards of the journal.

Thank you once again for your support and insightful feedback.

Best regards,

Minseong Kim